# Navigating Uncertainty: Managing Influenza-Associated Invasive Pulmonary Aspergillosis in an Intensive Care Unit

**DOI:** 10.3390/jof10090639

**Published:** 2024-09-07

**Authors:** Giacomo Casalini, Andrea Giacomelli, Laura Galimberti, Riccardo Colombo, Laura Milazzo, Dario Cattaneo, Antonio Castelli, Spinello Antinori

**Affiliations:** 1III Division of Infectious Diseases, ASST Fatebenefratelli-Sacco, Luigi Sacco Hospital, 20157 Milan, Italy; giacomo.casalini@asst-fbf-sacco.it (G.C.); andrea.giacomelli@unimi.it (A.G.); laura.galimberti@asst-fbf-sacco.it (L.G.); laura.milazzo@asst-fbf-sacco.it (L.M.); 2Department of Biomedical and Clinical Sciences, DIBIC, Università degli Studi di Milano, 20157 Milan, Italy; 3Anesthesia and Intensive Care Unit, ASST Fatebenefratelli-Sacco, Luigi Sacco Hospital, 20157 Milan, Italy; riccardo.colombo@asst-fbf-sacco.it (R.C.); antonio.castelli@asst-fbf-sacco.it (A.C.); 4Unit of Clinical Pathology, ASST Fatebenefratelli-Sacco, 20157 Milan, Italy; dario.cattaneo@asst-fbf-sacco.it

**Keywords:** invasive pulmonary aspergillosis, influenza, galactomannan antigen, IAPA, refractory disease, therapeutic drug monitoring

## Abstract

We present a challenging case of a patient admitted to an intensive care unit with influenza-associated pulmonary aspergillosis (IAPA). The clinical course was characterised by refractory fungal pneumonia and tracheobronchitis, suspected drug-induced liver injury due to triazole antifungals, and secondary bacterial infections with multidrug-resistant microorganisms, resulting in a fatal outcome despite the optimisation of antifungal treatment through therapeutic drug monitoring. This case underscores the complexity that clinicians face in managing critically ill patients with invasive fungal infections.

## 1. Introduction

Invasive pulmonary aspergillosis (IPA) is a life-threatening infection that is typically observed in patients with specific risk factors such as severe and prolonged neutropenia, frequently observed in haematological malignancies (HMs) and haematopoietic stem cell transplant (HSCT) recipients [1]. In recent years, IPA has been increasingly diagnosed in patients without these traditional risk factors, particularly those with chronic obstructive pulmonary disease (COPD), liver cirrhosis, or respiratory viral infections like influenza (influenza-associated pulmonary aspergillosis—IAPA) and COVID-19 (COVID-19-associated pulmonary aspergillosis—CAPA). IPA is the leading cause of death among invasive fungal infections and warrants the highest prioritisation for scientific research on a global scale [2,3,4].

IPA typically develops in patients with virus-induced acute respiratory distress syndrome (ARDS) who are admitted to intensive care units (ICUs) requiring mechanical ventilation. The clinical presentation is non-specific, involving fever, increase in respiratory secretion, and worsening of ventilatory parameters despite broad-spectrum antibiotic therapy. Sometimes, whitish plaques adhering to the bronchial mucosa can be observed, indicating tracheobronchitis. The typical radiological findings of IPA seen in neutropenic patients, such as nodules, the halo sign, cavitations, and the air-crescent sign, are rarely observed in critically ill patients, probably because the infection’s pathogenesis is different from the haematological patients and partly because performing computed tomography (CT) scans can be challenging with patients who have been intubated [5,6].

Diagnosis of IPA relies on examining samples from lower respiratory tract, primarily bronchoalveolar lavage (BAL), for microscopic evidence of hyphae, fungal culture, and the presence of the galactomannan antigen (GM). Fungal biomarkers, including GM and β-D-glucan (BDG), can also be searched for in serum, although the sensitivity is lower in non-neutropenic patients given that angioinvasion seem to be less frequent (at least in patients with CAPA) [6,7,8]. Clinical and research criteria have been developed to help clinicians recognise and diagnose IPA in critically ill patients; moreover, it is increasingly clear that diagnostic strategies, such as a routine screening surveillance in the ICU with fungal biomarkers in patients at risk, are crucial for an early diagnosis [2,6,9,10]. However, a common limitation of these criteria is their low ability to differentiate between true infection and airway fungal colonisation. Histological examination of biopsies is the gold standard for making this distinction, but it is rarely feasible in critically ill patients; therefore, most patients are diagnosed with probable or putative aspergillosis based on fungal biomarkers or cultures [11].

IPA in the ICU is associated with a significant increase in mortality, with rates as high as 50%. Treatment strategies are similar to those used in immunocompromised patients, with triazoles being used as the first-line treatment. However, triazoles have a narrow therapeutic index and are frequently associated with side effects, including drug-induced liver injury [12]. Pathophysiological changes in critically ill patients alter the pharmacokinetics (PK) of antifungals, making therapeutic drug monitoring (TDM) crucial for optimising antifungal treatment in the ICU. TDM helps achieve therapeutic concentrations while reducing the risk of side effects [13]. Finally, evaluating the clinical response in such patients can be challenging. Clinicians typically rely on fungal biomarkers and radiological follow-up, which are sometimes difficult to interpret due to underlying viral ARDS and other complications, particularly secondary bacterial infections. In this setting, distinguish refractory fungal disease from clinical deterioration caused by other factors can be very difficult.

Here, we report a complex case of probable IAPA managed in the ICU, which highlights common challenges encountered in clinical practice, such as the evaluation of refractory disease, how to manage refractory disease, drug-induced liver injury (DILI), PK variability, and secondary infections.

## 2. Case Report

This is the case of a 47-year-old Egyptian woman who sought medical attention for difficulty in breathing and fever. The patient had type 1 diabetes with poor glycaemic control, hypertension, and class I obesity (body mass index of 31 kg/m^2^). She had been living in Italy for 10 years and, at the end of November 2023, she travelled back to Egypt to visit her family. During her stay, she developed an influenza-like syndrome and was admitted to a regional hospital with a diagnosis of pneumonia and mild respiratory failure. She received broad-spectrum antibiotics (meropenem, amikacin), but her symptoms did not resolve. The medical records indicated a sputum culture positive for *Aspergillus fumigatus*, but other details on antifungal therapies were not available. The patient decided to self-discharge from the hospital, travelled back to Italy, and arrived at the Emergency Department (ED) of our hospital on December 14th.

Upon arrival at the ED, the patient had high grade fever and severe respiratory failure, with an oxygen saturation of 76% while breathing ambient air. Her blood tests revealed a white blood cell count of 29,010/µL and a C-reactive protein of 325 mg/L. A contrast-enhanced CT scan of the chest showed bilateral diffuse consolidations, and a nasopharyngeal swab tested positive for influenza A [polymerase chain reaction (PCR) test positive for H1N1] and negative for SARS-CoV-2 (both rapid antigen test and PCR). Despite increased oxygen delivery via mask, her clinical condition rapidly deteriorated, necessitating orotracheal intubation and mechanical ventilation in the ICU. Upon ED arrival, broad antimicrobial therapy with meropenem, vancomycin, and intravenous voriconazole (350 mg every 12 h on day 1, then 200 mg every 12 h) was started, along with oseltamivir for influenza. A single dose of intravenous methylprednisolone was administered. Bronchoscopy with BAL performed on ICU admission confirmed the diagnosis of probable or putative IAPA according to available definitions: the entry criterion of ICU admission because of viral ARDS, compatible signs and symptoms, bilateral pulmonary infiltrates, and positive mycological tests (BAL culture positive for *Aspergillus fumigatus*, BAL-GM of 5.6, and hyphae observed on BAL microscopy) [2,9,10,14]. The patient also exhibited tracheobronchitis, characterised by whitish plaques on the bronchial mucosa with easy mucosal bleeding (Figure 1).

During the ICU stay, the clinical course was characterised by persistent respiratory failure, poorly responsive even to prone positioning cycles, and recurrent high-grade fever. On a chest CT scan on day 7, bilateral consolidations had increased in extent, and by day 14, cavitations had appeared. On two consecutive follow-up bronchoscopies, BAL-GM levels increased despite treatment with adequately dosed voriconazole (serum levels were above the lower limit for efficacy, 1 mcg/mL) and subsequent combination therapy with caspofungin (70 mg every 24 h) starting from day 7 (Figure 2, Table 1). Tracheobronchitis showed no improvement, with only a few areas of healthy bronchial mucosa remaining, and microscopic examination of BAL fluid persistently revealed hyphae.

Based on full antifungal susceptibility of the cultured *Aspergillus fumigatus* strain (an E-test for voriconazole, posaconazole, itraconazole, isavuconazole, and liposomal amphotericin B was performed by the Medical Mycology Laboratory at the Università degli Studi di Milano), together with therapeutic levels of voriconazole, a diagnosis of refractory IAPA was feasible, defined as an “infection with worsening or new attributable clinical signs or symptoms or radiological findings while on antifungal treatment” [15].

Due to concurrent liver injury, characterised by a significant increase in serum bilirubin and gamma-glutamyl transferase (GGT) with normal aminotransferases, an abdominal ultrasound and CT scan were performed, and both were unremarkable. This led to a suspicion of DILI from triazole treatment (Figure 3). Voriconazole was replaced with intravenous isavuconazole (200 mg every 24 h, after the loading dose), and inhaled liposomal amphotericin B was introduced (day 13, 50 mg every 12 h), resulting in a mild improvement in respiratory exchanges.

However, this improvement was short-lived, and the patient’s condition subsequently worsened, with declining renal function and severe hypercapnia. Therefore, on day 17, continuous veno-venous hemofiltration (CVVH) and extracorporeal carbon dioxide removal were started. As a result, the isavuconazole dosage was doubled (200 mg every 12 h), with serum drug levels remaining within the therapeutic range (Table 1). Repeated chest CT scans on day 18 and 24 showed an increase in the magnitude of cavitations. In contrast, BAL-GM levels finally decreased (starting from day 22), although the culture specimen still tested positive for *A. fumigatus* (susceptibility test was not available). Due to a further worsening of liver injury without an apparent cause, isavuconazole was discontinued on day 29 and replaced with intravenous liposomal amphotericin B (L-AMB) on day 32 (350 mg every 24 h, i.e., 4 mg/Kg/day).

The clinical course was further complicated by concurrent infections, including ventilator-associated pneumonia (VAP) from *Escherichia coli* producing the carbapenemases NDM/OXA-48 (day 8, treated with ceftazidime/avibactam plus aztreonam for 7 days), uncomplicated central-line-associated candidemia from *Candida glabrata* (day 15, treated with caspofungin for 14 days), and VAP from *K. pneumoniae* producing the carbapenemases NDM/OXA-48 (day 17, treated with ceftazidime/avibactam plus aztreonam for 9 days). After just over a month of hospitalisation in the ICU, the patient passed away following another episode of septic shock with irreversible hypercapnic respiratory failure. 

## 3. Discussion

IAPA presents a significant challenge in patients with ARDS, requiring invasive mechanical ventilation, and our case report highlights some issues that are frequently encountered in clinical practice (i.e., define the response to treatment, DILI, PK/PD variables, and secondary bacterial infections).

Diagnosing invasive fungal infections in the ICU is challenging, in part due to the limitations in the performance of fungal biomarkers and the difficulties in obtaining appropriate specimens due to the instability of patients. According to current definitions, our patient had probable/putative IAPA plus tracheobronchitis [2,9,10,14]. The entry criterion was satisfied, as the patient required ICU admission for ARDS with a positive influenza PCR test temporally related to ICU admission. Bilateral pulmonary infiltrates were noted on the chest CT scan, thus meeting the radiology criterion. Regarding the mycological criterion, the BAL culture was positive for *Aspergillus fumigatus*, GM was positive with an ODI of 5.6, and microscopic examination revealed acute angle, dichotomous branching, and septate hyphae. Of note is that the serum GM was repeatedly negative (reference range: <0.5 ODI). Additionally, the patient had tracheobronchitis, presenting as whitish plaques adhering to the bronchial mucosa and easy mucosal bleeding. However, we must note that IAPA was not confirmed with a biopsy from the tracheal or lung lesion due to the patient’s clinical instability and risk of bleeding throughout the entire hospitalization. The autopsy was not performed due to the lack of consent from their relatives; therefore, a definitive diagnosis could not be reached. According to different guidelines, the first-line treatment for invasive pulmonary aspergillosis is a mould-active triazole (voriconazole or isavuconazole) [6,10,12,16]. Voriconazole is the most used drug in clinical practice, and it was chosen as the primary treatment in our case, also considering the susceptibility profile of the first *A. fumigatus* isolated. Following two weeks of mould-active treatment and despite adequate drug exposure the patients experienced a clinical and radiological progression together with any mycological response (Table 1). After ruling out a concomitant fungal infection and a possible immune reconstitution syndrome (IRIS), we diagnosed a refractory IPA [15].

The first strategy is to switch the triazole, considering isavuconazole or posaconazole according to the fungal susceptibility profile (if available); both drugs showed their non-inferiority towards voriconazole in randomised clinical trials (RCTs) together with a lower risk of drug–drug interactions, more predictable PK, and fewer side effects, as we usually see in the clinical practice [12,16,17,18]. The second option is to add an echinocandin to the triazole drug, as these antifungals have different targets in fungal cells (inhibition of beta-D-glucan and interruption of ergosterol synthesis, respectively) and could potentially be synergistic. However, in vitro studies indicate that most combinations either show strong synergy only in few isolates or work across a broad range of isolates with only weakly synergistic interactions [19]. Minimal clinical data are available to support routine combination therapy. In addition to two small observational studies that showed improved survival in haematological patients with IPA treated with voriconazole plus caspofungin compared to liposomal amphotericin B, a landmark RCT from 2015 involving patients with hematologic malignancies or haematopoietic cell transplantation demonstrated a significant reduction in mortality for those receiving the combination therapy for at least two weeks [20,21,22]. The benefit was even greater if GM was positive in either serum or BAL, resulting in a mortality reduction of 11.5 percentage points (confidence interval [CI], −22.7 to −0.4; *p* = 0.037) [20,21,22]. In our case, we employed both strategies: adding caspofungin and switching from voriconazole to isavuconazole. This decision was made because of the simultaneous increase in both total bilirubin and GGT, which raised suspicions of voriconazole-induced liver injury. The subsequent clinical course showed an improvement in mycological parameters, with a decrease in BAL GM levels (from 7.1 to <1.0 ODI), although *A. fumigatus* culture from BAL was still positive on day 22 (unfortunately, we were not able to perform the susceptibility test because it is not readily available in our centre). Isavuconazole plasma levels were within therapeutic range, even slightly increased at first (Table 1). 

The clinical scenario was further complicated by the fungal tracheobronchitis which worsened over time. Systemic drugs alone may be inadequate due to limited antifungal penetration and poor accessibility to the bronchi [23]. In such cases, adding nebulised antifungals (i.e., L-AMB) to systemic therapy could be beneficial, although clinical evidence supporting nebulised therapy is limited to case reports and single-centre case series [24,25,26]. One major concern with this approach is about the delivery system used for the antifungal, which must ensure that optimal concentrations of the drug reach the bronchi without adhering to the ventilation system [27]. In our case L-AMB was delivered using an ultrasonic nebulizer and it was diluted in normal sterile saline solution. However, the ventilation system frequently malfunctioned due to the occlusion of the respiratory membranes. 

Salvage therapy in case of further disease progression involves switching to L-AMB, particularly if the patient is unstable or experiencing rapid disease progression; an echinocandin can be used in combination, once again considering the different therapeutic target of these two antifungal classes [12,15,16]. The combination of L-AMB and caspofungin has been evaluated in two small observational studies on patients with HM, either as initial therapy or salvage regimen, showing a little improvement in clinical outcomes [28,29]. It should be highlighted that the combination of amphotericin B and triazoles is generally not recommended because of possible antagonisms. Finally, L-AMB should be the first choice in cases of a local high prevalence of azole resistance among *Aspergillus* species [15]. In our case, the triple combination of isavuconazole, caspofungin (re-introduced on day 15 because of candidemia), and inhaled L-AMB resulted in a partial response regarding the fungal infection: BAL-GM was negative in the last sample, but hyphae were still present on cytology, along with tracheobronchitis plaques suggesting that IAPA was not sufficiently controlled. 

Proven triazole resistance in *Aspergillus* spp. can cause refractory disease, both as primary IPA and as breakthrough infections during triazole treatment. Resistance rates vary significantly by country, reaching up to 20% in the Netherlands and Belgium, though surveillance data are not readily available for all regions [30]. In Italy, recent epidemiological studies report triazole resistance rates in *Aspergillus fumigatus* ranging from 0.8% to 6.6% [31,32,33]. Triazole-resistant IPA is associated with increased mortality, likely due to delays in diagnosis and appropriate treatment [30]. In our case, primary antifungal resistance was ruled out in the first positive BAL culture. However, susceptibility testing could not be performed on the second positive BAL culture due to logistical issues, so the possibility of resistance emerging during treatment as a cause of refractory disease cannot be excluded, as noted in a previous report [34]. Pathophysiological changes in ICU patients induced by the underlying disease and the treatment administered can alter the PK of antimicrobial agents (especially in terms of volume of distribution and clearance), potentially leading to subtherapeutic levels or, conversely, toxic levels that increase the risk of side effects [35]. Voriconazole TDM is strongly recommended by guidelines, especially in critically ill patients, considering the inter-individual PK variability of voriconazole levels; the target through level is 1.5–5 μg/mL [36,37]. In our patient plasma voriconazole levels where above the minimum concentration (1 μg/mL) associated with improved clinical outcomes [13,36]. On day 17, the patient was started on CVVH due to worsening renal failure with anuria, while concurrently receiving isavuconazole and caspofungin. Given its high protein binding, isavuconazole is theoretically not significantly dialysed and exhibits less inter-individual PK variability compared to voriconazole. Target through levels for isavuconazole are 1–5 μg/mL [36]. Subtherapeutic concentrations have been observed during continuous hemofiltration, meaning that TDM is strongly recommended [38,39]. According to a French study, the optimal trough concentration should be at least 2 μg/mL [40]. In our patient, CVVH started on day 3 of isavuconazole treatment, initially given as 200 mg every 24 h after the loading dose. Four days after starting CVVH (day seven of treatment), the isavuconazole dose was empirically increased to 200 mg every 12 h. TDM conducted the next day showed elevated serum levels (6 μg/mL). Six days later, serum levels returned to the normal reference range, confirming the efficacy of the empirical dose adjustment (Table 1). Echinocandins exhibit high plasma protein binding (≥97–99%), and in critically ill patients, exposure is typically lower and highly variable [41]. When TDM is performed, peak levels (C_max_) are measured and a target C_max_/MIC ratio of at least 10 mg/L is desirable. In our patient caspofungin was administered from day 15 at a dose of 70 mg every 24 h. Four days after the initiation of CVVH, C_max_ was 8.7 mg/L. Consequently, the dose was increased to 100 mg every 24 h, resulting in a doubling of serum levels after 4 days (C_max_ 16.1 mg/L).

Patients with IFI in the ICU often have poor outcomes. IAPA is linked to more severe disease progression, higher complication rates, longer ICU stays, and increased need for organ support. A recent meta-analysis of IAPA in critically ill patients showed a clear association with ICU mortality (odds ratio [OR] 2.6; 95% CI 1.8–3.8) [5]. Similar findings were reported for CAPA, with mortality exceeding 50% [42]. In clinical practice, it is challenging to determine whether clinical failure and mortality are driven by IFIs or by viral ARDS (influenza or COVID-19-related), secondary infections, and other organ failures. Autopsy studies of ICU patients with IFI showed mainly sparsely scattered, fragmented hyphae in acutely inflamed and/or necrotic lung tissue with little or no angioinvasion. This suggests that the main driver of lung failure and poor outcome might be the overwhelming inflammatory response triggered by the initial viral infection [11,43,44,45] Lastly, the impact of secondary infections on poor outcomes is well recognised, especially those caused by multidrug-resistant microorganisms, which substantially elevate the risk of mortality in these patients [46]. In our case, the occurrence of two episodes of VAP and candidemia undoubtedly had a profound impact on the outcome and, ultimately, the patient likely succumbed to septic shock. Given the complex clinical scenario and overlapping infections, it is plausible that the clinical outcome was primarily influenced by these secondary infections rather than the presumed refractory IPA.

## Figures and Tables

**Figure 1 jof-10-00639-f001:**
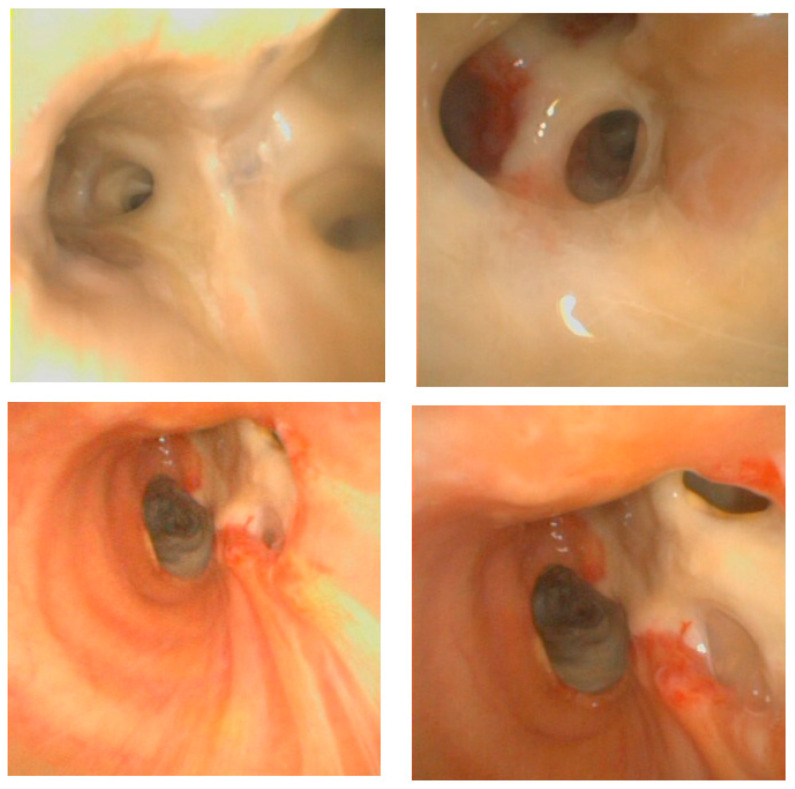
Whitish plaques adhering to the bronchial mucosa, consistent with tracheobronchitis.

**Figure 2 jof-10-00639-f002:**
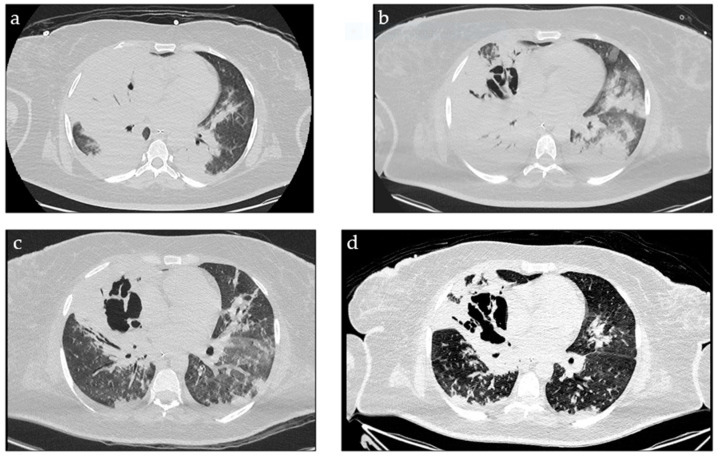
Pulmonary lesions on repeated CT scans: (**a**) day 7; (**b**) day 14; (**c**) day 18; (**d**) day 24.

**Figure 3 jof-10-00639-f003:**
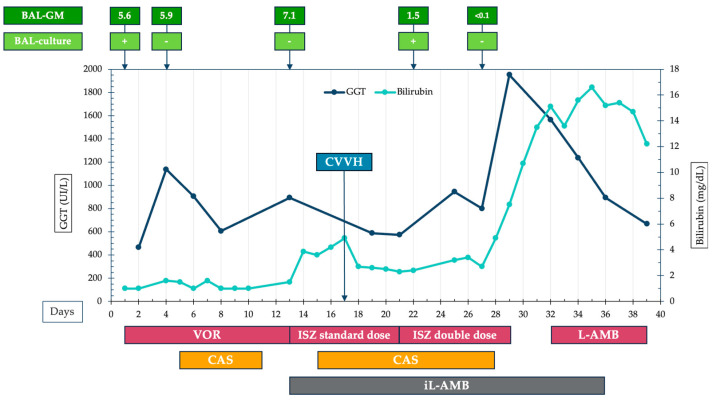
Timeline of the antifungal treatment with trend values of gamma-glutamyl transferase (GGT) and total bilirubin levels. VOR, voriconazole; ISZ, isavuconazole; L-AMB, liposomal amphotericin B; iL-AMB, inhaled L-AMB; CAS, caspofungin; CVVH, continuous veno-venous hemofiltration; BAL, bronchoalveolar lavage; GM, galactomannan.

**Table 1 jof-10-00639-t001:** Bronchoalveolar lavage data and serum drug levels.

	D1	D4	D6	D13	D21	D22	D25	D27	D28	D33
BAL cytology	Hyphae	Hyphae		Hyphae				Hyphae		
BAL-GM (ODI)	5.6	5.9		7.1		1.5		<1.0		
BAL-culture	+	−		−		+		−		
VOR (mcg/mL), through levels			3.8	2.2						
ISZ (mcg/mL), through levels						6			1.9	1.4
CAS (mg/L), peak levels					8.7		16.1			

**Abbreviations**: BAL, bronchoalveolar lavage; GM, galactomannan antigen; ODI, optical density index; VOR, voriconazole; ISZ, isavuconazole; CAS, caspofungin. BAL culture was positive (+) for *A. fumigatus* both on D1 and D22 and negative (−) on D4-13-27. Reference range for VOR and ISV serum levels: 1–5 μg/mL. Reference range for CAS serum levels: C_max_/MIC at least 10 mg/L.

## Data Availability

The original contributions presented in the study are included in the article, further inquiries can be directed to the corresponding author.

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
