# Peer review of "Navigating Uncertainty: Managing Influenza-Associated Invasive Pulmonary Aspergillosis in an Intensive Care Unit"

_jof, 2024, doi:10.3390/jof10090639_

Round 1

Reviewer 1 Report

Comments and Suggestions for Authors

This article is well-written and presents the difficulties in managing influenza-associated pulmonary aspergillosis in the ICU setting. Some comments are listed below:

-    Were all voriconazole and isavuconazole administrated intravenously?

-    Line 138: It is not clear what is meant by “Based on full antifungal susceptibility of the cultured Aspergillus fumigatus strain…”. Did it mean that the A. fumigatus isolate was susceptible to amphotericin B, voriconazole, isavuconazole, posaconazole, and caspofungin determined by MICs based on antifungal susceptibility testing?

-    Line 161, please provide the L-AmB dose in mg/kg/day.

-    It is not clear in lines 201-205, which mention "Clinical data to support routine combination therapy are lacking." This is because the following sentences (lines 205-211) indicate better clinical efficacy with combination therapy compared to monotherapy in three studies, and in two studies, combination therapy was used as primary therapy.

-    Did the patient ever received steroids during her ICU stay?

-    The BAL culture turned out to be positive for A. fumigatus again. There is a possibility that the patient acquired an azole-resistant A. fumigatus during the later stage.

-    The author may consider moving the reference bracket [19-21] in line 211 to line 209 at the end of the sentence ("...at least two weeks [19-21]") and leaving a specific reference for lines 210-211 ("…(CI, -22.7 to -0.4; P = 0.037) [21]"), if correct.

-    Table 1: The author might consider including the caspofungin serum level (mentioned in line 269) in Table 1.

-    Figure 3: The authors may consider adding a note indicating the date when CVVH started and separating the isavuconazole timetable into original doses and double doses for better readability.

Author Response

Reviewer #1

This article is well-written and presents the difficulties in managing influenza-associated pulmonary aspergillosis in the ICU setting. Some comments are listed below:

-    Were all voriconazole and isavuconazole administrated intravenously?

Yes, the patient was intubated and sedated, and all the drugs were administered intravenously, except for inhaled liposomal amphotericin B.

-    Line 138: It is not clear what is meant by “Based on full antifungal susceptibility of the cultured Aspergillus fumigatus strain…”. Did it mean that the A. fumigatus isolate was susceptible to amphotericin B, voriconazole, isavuconazole, posaconazole, and caspofungin determined by MICs based on antifungal susceptibility testing?

Yes, the A. fumigatus strain isolated from the BAL sample showed full susceptibility to all tested antifungals (see below).

-    Line 161, please provide the L-AmB dose in mg/kg/day.

The dose was 4 mg/Kg/day, the full test was updated.

-    It is not clear in lines 201-205, which mention "Clinical data to support routine combination therapy are lacking." This is because the following sentences (lines 205-211) indicate better clinical efficacy with combination therapy compared to monotherapy in three studies, and in two studies, combination therapy was used as primary therapy.

We thank the Reviewer for this comment. In lines 201-205, we intended that there is limited data supporting routine combination therapy - specifically, only two observational studies and one small RCT, as described in lines 205-211. We have revised the sentence based on the suggested changes.

-    Did the patient ever received steroids during her ICU stay?

The patient received systemic steroids (specifically, IV methylprednisolone) only upon arrival at the emergency department due to bronchospasm.

-    The BAL culture turned out to be positive for A. fumigatus again. There is a possibility that the patient acquired an azole-resistant A. fumigatus during the later stage.

Thank you to the reviewer for this interesting question. This is a possible scenario; however, we were unable to perform an antifungal susceptibility test on the second positive BAL samples, as these tests are not routinely conducted in our laboratory (we typically send samples to the regional reference laboratory, but this was not performed for the second positive BAL).

-    The author may consider moving the reference bracket [19-21] in line 211 to line 209 at the end of the sentence ("...at least two weeks [19-21]") and leaving a specific reference for lines 210-211 ("…(CI, -22.7 to -0.4; P = 0.037) [21]"), if correct.

We thank the reviewer for the suggestion; the manuscript was modified accordingly.

-    Table 1: The author might consider including the caspofungin serum level (mentioned in line 269) in Table 1.

The table was updated according to the suggestion.

-    Figure 3: The authors may consider adding a note indicating the date when CVVH started and separating the isavuconazole timetable into original doses and double doses for better readability.

The figure was updated according to the suggestion.

Reviewer 2 Report

Comments and Suggestions for Authors

Comments to the authors,

The authors described a complex case of IAPA. IAPA and CAPA are emerging issue in the management of influenza and COVID-19, respectively, and need more attention and research. 

Major comments:

1. The patient suffered from influenza-like illness in Egypt from November to December 2023. After the admission to the authors' hospital on December 14, a diagnosis of influenza was made by NAAT. However no testing for SARS-CoV-2 was mentioned. SARS-CoV-2 should be considered as one of the major pathogens of influenza-like illness, and the testing for SARS-CoV-2 infection, either by rapid antigen test or NAAT, should be performed in any case of ILI together with influenza test. Influenza virus and SARS-CoV-2 have been shown to cause co-infection. The authors should show the result of SARS-CoV-2 test, and make it clear that the pathophysiology of this patient was not related to CAPA, but only affected by IAPA.

2. The empirical administration of voricolazole was not effecteve. Even though the authors state that the cultured Aspergillus fumigatus was fully susceptible to antifungals, the result of antifungal susceptibility testing should be shown. In addition the authors should discuss the azole-resistant Aspergillosis as a reason of refractory pulmonary aspergillosis. Consider the reference; https://academic.oup.com/jid/article/216/suppl_3/S436/4107054

3. Consider the addition of the recently-published reference; https://www.thelancet.com/journals/lanres/article/PIIS2213-2600(24)00151-6/fulltext.

MInor comment:

1. The word "routinary" (line 204) would be better to be replaced by "routine".

Comments on the Quality of English Language

Only minor errors were found.

Author Response

Reviewer #2

The authors described a complex case of IAPA. IAPA and CAPA are emerging issue in the management of influenza and COVID-19, respectively, and need more attention and research. 

Major comments:

  1. The patient suffered from influenza-like illness in Egypt from November to December 2023. After the admission to the authors' hospital on December 14, a diagnosis of influenza was made by NAAT. However, no testing for SARS-CoV-2 was mentioned. SARS-CoV-2 should be considered as one of the major pathogens of influenza-like illness, and the testing for SARS-CoV-2 infection, either by rapid antigen test or NAAT, should be performed in any case of ILI together with influenza test. Influenza virus and SARS-CoV-2 have been shown to cause co-infection. The authors should show the result of SARS-CoV-2 test, and make it clear that the pathophysiology of this patient was not related to CAPA, but only affected by IAPA.

Thank you to the author for the comment. SARS-CoV-2 was tested using both a rapid antigen test and NAAT on nasopharyngeal swab and BAL samples, and the results were negative. The main text has been updated to include this information.

  1. The empirical administration of voriconazole was not effective. Even though the authors state that the cultured Aspergillus fumigatus was fully susceptible to antifungals, the result of antifungal susceptibility testing should be shown. In addition the authors should discuss the azole-resistant Aspergillosis as a reason of refractory pulmonary aspergillosis. Consider the reference; https://academic.oup.com/jid/article/216/suppl_3/S436/4107054

We thank the reviewer for this comment. The antifungal susceptibility test results for A. fumigatus are shown here; we chose not to include them in the main text to prioritize other figures and tables. A discussion on azole resistance has been added.

  1. Consider the addition of the recently-published reference; https://www.thelancet.com/journals/lanres/article/PIIS2213-2600(24)00151-6/fulltext.

We thank the reviewer for suggesting this reference. However, most of the data reported in this review can be found in the citations already present in the main text, so we decided not to include this reference.

MInor comment:

  1. The word "routinary" (line 204) would be better to be replaced by "routine".

The main text was corrected according to the suggestion.

Reviewer 3 Report

Comments and Suggestions for Authors

This report of a possible Aspergillus tracheobronchitis and pneumonia is not interpretable. It is regretable that no biopsy. of the tracheal lesion was taken and no autopsy performed. The patient could have had bronchopulmonary colonization, accounting for the culture and smear, and not aspergillosis. The discussion should have noted this possibility, as well as the tentative nature of the diagnosis. Without confirmation of the diagnosis, there is little that can be learned from this case. A minor issue is that the report should have indicated whether corticosteroids had been administered in Egypt, if that is known, and in Italy, because corticosteroids are often administered to patients in this situation. 

Author Response

Reviewer #3

This report of a possible Aspergillus tracheobronchitis and pneumonia is not interpretable. It is regretable that no biopsy. of the tracheal lesion was taken and no autopsy performed. The patient could have had bronchopulmonary colonization, accounting for the culture and smear, and not aspergillosis. The discussion should have noted this possibility, as well as the tentative nature of the diagnosis. Without confirmation of the diagnosis, there is little that can be learned from this case. A minor issue is that the report should have indicated whether corticosteroids had been administered in Egypt, if that is known, and in Italy, because corticosteroids are often administered to patients in this situation. 

Thank you to the reviewer for this comment. Diagnosing invasive fungal infections in the ICU is challenging due to the performance limitations of fungal biomarkers and the difficulties in obtaining biopsy samples, as ventilated patients are often unstable, and performing a biopsy could pose a significant risk to the patient. Therefore, the diagnosis is usually made using a combination of clinical, radiological, and mycological criteria, along with a host risk factor (in this case, influenza A). In lines 96-97, we clearly state that the patient had probable or putative IAPA according to available definitions. Additionally, a recent position paper from various Scientific Societies outlines the research definition for probable invasive aspergillosis and tracheobronchitis, and those definitions are definitely appliable to our case (Bassetti M, Giacobbe DR, Agvald-Ohman C, et al. Invasive Fungal Diseases in Adult Patients in Intensive Care Unit (FUNDICU): 2024 consensus definitions from ESGCIP, EFISG, ESICM, ECMM, MSGERC, ISAC, and ISHAM. Intensive Care Med. 2024;50(4):502-515. doi:10.1007/s00134-024-07341-7). This reference was included in the main text. Furthermore, the presence of tracheobronchial ulceration, nodules, pseudomembranes, plaques, or eschar on bronchoscopy is sufficient to diagnose probable tracheobronchitis. However, we have noted in the discussion that biopsies were not performed due to clinical instability.

Finally, the patient received systemic steroids (specifically, IV methylprednisolone) only upon arrival at the emergency department due to bronchospasm.

Round 2

Reviewer 2 Report

Comments and Suggestions for Authors

Comment to the authors,

The initial treatment by voriconazole was unsuccessful even though the isolated Aspergillus fumigatus strain was sensitive to voriconazole. Even though primary antifungal resistance was ruled out, there remained the possibility that the resistance to vorizonazole had emerged during the treatment by voriconazole, especially in this case that the susceptibility testing on the second isolate was not performed. Emergence of voriconazole resistance by amino acid substitution has been reported previously (Med Mycol. 2010 Feb;48(1):197-200.; J Clin Microbiol. 2012 Jul;50(7):2531-4.). The authors should discuss this point in Discussion.

Author Response

The initial treatment by voriconazole was unsuccessful even though the isolated Aspergillus fumigatus strain was sensitive to voriconazole. Even though primary antifungal resistance was ruled out, there remained the possibility that the resistance to vorizonazole had emerged during the treatment by voriconazole, especially in this case that the susceptibility testing on the second isolate was not performed. Emergence of voriconazole resistance by amino acid substitution has been reported previously (Med Mycol. 2010 Feb;48(1):197-200.; J Clin Microbiol. 2012 Jul;50(7):2531-4.). The authors should discuss this point in Discussion.

**We appreciate the reviewer's comment and agree that triazole resistance may have emerged during treatment. We have included this point in the discussion and updated the reference list accordingly.**

Reviewer 3 Report

Comments and Suggestions for Authors

It would have been helpful to know if the patient had received corticosteroids at the outside hospital and if she had a history of cigarette smoking. Steroids might have predisposed her to invasive aspergillosis and smoking to colonization. It is understandable that those details are not available.

The authors are correct that a patient like theirs is often treated for invasive aspergillosis even though the patient might have only bronchial colonization and no benefit from antifungal therapy. Publication requires a higher degree of certitude in order not to be misleading. The failure of the bronchoscopist to biopsy the lesion is regrettable from the point of view of management and publication but the fact is that the reader cannot conclude that the patient had invasive aspergillosis based on bronchoscopic appearance of a lesion and fungus in the respiratory tract.

Author Response

Comment 1: It would have been helpful to know if the patient had received corticosteroids at the outside hospital and if she had a history of cigarette smoking. Steroids might have predisposed her to invasive aspergillosis and smoking to colonization. It is understandable that those details are not available.

Response 1: History of cigarette smoking was not reported in the medical record, as well as corticosteroids treatment during the hospitalisation in Egypt.

The authors are correct that a patient like theirs is often treated for invasive aspergillosis even though the patient might have only bronchial colonization and no benefit from antifungal therapy. Publication requires a higher degree of certitude in order not to be misleading. The failure of the bronchoscopist to biopsy the lesion is regrettable from the point of view of management and publication but the fact is that the reader cannot conclude that the patient had invasive aspergillosis based on bronchoscopic appearance of a lesion and fungus in the respiratory tract.

Response 2: We agree with the reviewer that the ideal approach would have been to confirm the diagnosis in vivo through a biopsy of tracheal or pulmonary lesions or post-mortem via autopsy. However, as noted in the text, a biopsy was not feasible due to the patient's clinical instability and risk of bleeding (sometimes even BAL can be challenging in case of severe respiratory failure and bronchospasm), and we did not have consent to perform an autopsy. This clinical scenario is common in the ICU, highlighting the importance of clinical algorithms in supporting diagnosis. In the main text, we clearly state that pulmonary and bronchial aspergillosis were deemed 'probable' based on the available clinical/mycological data and definitions. We also acknowledge this as a limitation in our case report.

Round 3

Reviewer 3 Report

Comments and Suggestions for Authors

I agree with the authors that this patient might have had aspergillus tracheobronchitis based on the appearance. I think it is unlikely but possible the patient's pneumonia was due to invasive aspergillosis. There hundreds of cases in the literature just like this, with an excellent report in 2021 of 20 cases, review of the literature and reasoned argument that aspergillosis can be invasive in patients with severe influenza or coronavirus pneumonia (PMC 7774554). There are also autopsies series failing to find invasive aspergillosis in such patients (PMC1038307). What this case adds is just another case of possible aspergillosis in a patient with severe viral pneumonia. What the literature needs after hundreds of possible cases is a few cases of proven invasive aspergillosis. If an autopsy or biopsy of their patient had been performed, they might have had such a case. Circumstances beyond their control deprived them of that possibility.